# Measuring novice-expert sense of place for a far-away place: Implications for geoscience instruction

Anne U. Gold[1]*, Emily M. Geraghty Ward[1], Casey L. Marsh[1], Twila A. Moon[2], Spruce W. Schoeneman[3], Alia L. Khan[4], Megan K. Littrell[1]

1 Cooperative Institute for Research in Environmental Sciences (CIRES), CIRES Education and Outreach Program, University of Colorado Boulder, Boulder, Colorado, United States of America, 2 Cooperative Institute for Research in Environmental Sciences (CIRES), National Snow and Ice Data Center, University of Colorado Boulder, Boulder, Colorado, United States of America, 3 Environmental Sciences Department, Spruce W. Schoenemann, University of Montana Western, Dillon, MT, United States of America, 4 Department of Environmental Science, Western Washington University, Bellingham, WA, United States of America

* anne.u.gold@colorado.edu

**Data Availability Statement:** All relevant data are within the paper and its Supporting Information files.

## Abstract

Individuals usually develop a sense of place through lived experiences or travel. Here we introduce new and innovative tools to measure sense of place for remote, far-away locations, such as Greenland. We apply this methodology within place-based education to study whether we can distinguish a sense of place between those who have visited Greenland or are otherwise strongly connected to the place from those who never visited. Place-based education research indicates that an increased sense of place has a positive effect on learning outcomes. Thus, we hypothesize that vicarious experiences with a place result in a measurably stronger sense of place when compared to the sense of place of those who have not experienced the place directly. We studied two distinct groups; the first are people with a strong Greenland connection (experts, n = 93). The second are students who have never been there (novices, n = 142). Using i) emotional value attribution of words, ii) thematic analysis of phrases and iii) categorization of words, we show significant differences between novice's and expert's use of words and phrases to describe Greenland as a proxy of sense of place. Emotional value of words revealed statistically significant differences between experts and novices such as word power (dominance), feeling pleasantness (valence), and degree of arousal evoked by the word. While both groups have an overall positive impression of Greenland, 31% of novices express a neutral view with little to no awareness of Greenland (experts 4% neutral). We found differences between experts and novices along dimensions such as natural features; cultural attributes; people of Greenland; concerns, importance, or interest in and feeling connected to Greenland. Experts exhibit more complex place attributes, frequently using emotional words, while novices present a superficial picture of Greenland. Engaging with virtual environments may shift novice learners to a more expert-like sense of place, for a far-away places like Greenland, thus, we suggest virtual field trips can supplement geoscience teaching of concepts in far-away places like Greenland and beyond.

**Funding:** National Science Foundation awards OPP-2021503, OPP-2021275 and OPP-2021543 National Oceanic and Atmospheric Administration award NA17OAR4320101. The funders had no role in study design, data collection and analysis, decision to publish, or preparation of the manuscript.

**Competing interests:** The authors have declared that no competing interests exist.

## Introduction

The warming Arctic is causing rapid changes in the natural and socio-cultural environment of Greenland and other polar regions [1,2]. A changing climate accelerates these changes in the Arctic natural environment, and bold and fast reductions in the global emission of greenhouse gases are critical to slow or mitigate changes that are dramatically changing polar regions [3]. Education plays a central role in building support for policies that reduce greenhouse gas emissions and inspire behavioral change. Prior research suggests that a place-based teaching approach can result in effective learning of scientific concepts and phenomena [4,5], pro-environmental behaviors, attitudes, and awareness [6], and can inspire learners to engage in mitigation of climate change impacts [7–12].

In the Geosciences, places are commonly defined by their physical attributes, location on Earth and the processes that shape them [13]. In contrast, the social construct of "place" describes the relationship between a physical location and people's affective response as well as the social values and norms they associate with it [14–16]. Thus, a person's sense of place is defined through the relationships and emotions that connect them to a location [17,18]. Geographically, there is no limitation to the size and characteristics of a place that people are emotionally connected to, it can be vast or locally restricted [19,20].

Place-based science education centers instruction on places to achieve positive learning outcomes, often focusing on locations close to home to build on learners' own experiences and resulting sense of place [4]. The theoretical frameworks that underpin place-based science education connect science concepts to place-based knowledge [4,21–23] or other ways of knowing [24], providing context and relevance to concepts, phenomena or processes that may otherwise appear abstract. Research on place-based learning demonstrates that learning outcomes are more robust and lasting when the content is made relevant through the connection to one's community or a meaningful place [8–11,25]. Sense of place is usually developed for locations that hold personal meaning and that people have experienced directly, through lived experiences or visiting [22,25,26] but can be developed for far-away, remote places and through vicarious experiences [4]. Vicarious experiences of a place include virtual field trips [27,28] and exposure to place-related information through stories or other media representations [29]. For example, the tourism or location marketing industries craft deliberate destination images and evoke place-based feelings in their advertisement campaigns to construct their ideal image of a place [30–32].

Geoscience educators often teach scientific concepts using examples from field sites at locations far away, many of which students have not visited before, and may never visit. These locations are usually selected because scientific phenomena or processes can be readily observed at these sites. Visiting field sites is an important experience to connect with and learn about places and the processes that shape them [33]. Studying the potential difference between sense of place of those who have strong connections to a place versus novices who have no vicarious experience with the place will provide important insights into geoscience education, both for field education and when using examples of field locations during instructions. Future studies may explore if virtual field trips [34–42] may result in an improved sense of place and thus be a valuable addition to instruction in the geosciences.

Here we present new approaches to measure the sense of place that people hold for Greenland—an example of a remote, far-away location. Using tools from cognitive psychology research, we analyze words and phrases that people use to describe Greenland as a means to express their sense of place. We know from linguistics research that language is an important vehicle to share feelings and thoughts and often conveys affect (e.g., emotions, sentiment, attitudes). Words play an important role in our understanding and description of the world

around us [43], either explicitly through their core meaning or implicitly through connotation [44]. To generate a baseline understanding of the sense of place that people hold, we surveyed geoscience students who have never visited Greenland and compared these findings to the results from people who either live in or have conducted field research in Greenland. We explored the research questions: *In what ways does the sense of place of novices (people who have never visited Greenland before) compare to the sense of place of experts (people who have spent time in Greenland)*? We quantified elements of a person's sense of place and measured the impacts of a more expert-like sense of place on learning outcomes.

## Methodology

The goals for this study were i) the development of a methodology to analyze keywords and short phrases that survey respondents used to describe their associations with Greenland to infer their sense of place for Greenland and ii) the development of a comparison set of benchmarks of what sense of place looks like for a group of people who have never been to Greenland (novices) compared to those who have spent time in Greenland (experts).

### Theoretical framework for the study design

Research results demonstrate that sense of place is a multidimensional construct that is both shaped by the meaning a place has to people (place meaning, [45,46]) and peoples' emotional attachment to a particular place (place attachment, [15,20,47]). Place meaning is the symbolic interpretation that people associate with or ascribe to a place and is often grounded in experiences such as shared histories, stories, challenges, or politics of a place [24]. Thus, place meaning varies for individuals or communities and depends on their experience with a place and can be expressed through historical, scientific, aesthetic, or cultural meaning [48,49]. Place attachment describes the emotional bond that individuals have with a specific location. It includes the place itself, the person experiencing the attachment and the cognitive, affective or behavioral connections formed through experiences with the place [20,49]. For this study, we characterize the sense of place for two distinct populations (novices and experts) by eliciting the meaning they hold for Greenland (Fig 1). As with other studies of sense of place, we view place meaning as the foundation for place attachment (e.g., [50]). Thus, place attachment can be considered an outcome of place meaning [51–53]. This theoretical framework informed our approach to both the collection and the analysis of the data in order to characterize sense of place.

### Survey instrument

A brief online survey was developed to measure the sense of place that respondents hold about Greenland (see S1 File). The open-ended survey prompts were based on the place meaning work of [30] and were focused on gathering information about the participants' feelings for and descriptions of Greenland. To develop his instrument, Young generated a list of place meaning items that characterized his study location, an Australian tourist destination. Based on visitor interviews and coding of information materials about the location, he distilled single meaning items (e.g., pristine) to describe the different facets of place meaning. In our survey, each respondent was asked two questions about Greenland. The first question: "*In no more than two sentences, how do you personally feel about Greenland*?" was slightly modified from one of Young's [30] interview questions to be specific to Greenland. In asking respondents to keep their answer to two sentences, we forced sharing of the initial associations with the place and allowed for comparable response length between study subjects. The second survey question "*What words would you use to describe Greenland*?" was inspired by the list of single place

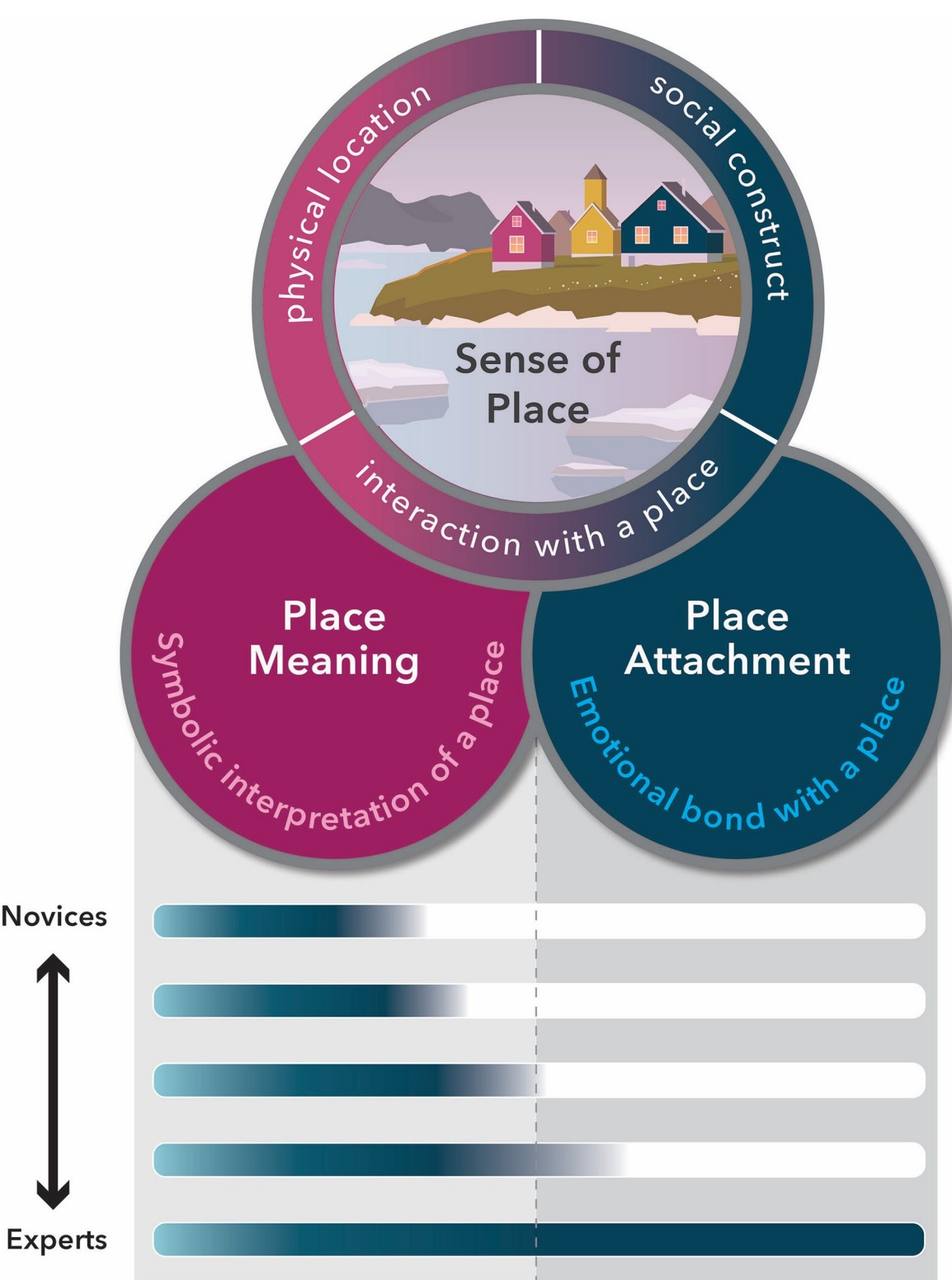

**Fig 1. Theoretical framework illustrating how the subconstructs place meaning and place attachment are related to sense of place.** The bar graphs summarize the findings from this study.

meaning items that are contained in the place meaning instrument that [30] developed based on his analysis. The survey further included two demographic questions asking about the respondent's age and profession. We tested the validity of the questions through a think-aloud interview with five study subjects. No adjustments were made after the test run. All participants provided electronic consent to participate in the study. We received IRB approval for this study from the University of Colorado IRB office under protocol 21–0189.

## Study participants

Novice participants were students recruited from undergraduate science courses at the University of Colorado Boulder through invitation by the course instructors. We selected U.S. undergraduate college students as our novice population because very few students at a U.S. college would have visited Greenland before, thus, responding to the survey with the expertise of a novice. The reason why we surveyed undergraduate students is because this study is part of a larger study that aims to improve undergraduate college instruction using immersive virtual field experiences and therefore, responses from college students provide a meaningful baseline for novices. Experts were recruited as a convenience sample via email, through email lists, online messaging boards and social media in Arctic or Greenland-focused networks and through professional organizations and agencies (such as US Interagency Arctic Research Policy Committee and Association of Polar Early Career Scientists). These networks include many people who are conducting research in Greenland, live in Greenland or spent time in Greenland, and thus provide access to people who have visited, worked, lived or currently live in Greenland. The survey included a question about the respondent's relationship to Greenland to determine whether a respondent was counted as an expert or a novice.

We collected 142 responses from novices. A total of 102 experts completed the survey, but nine respondents indicated that they had never been to Greenland and thus were excluded from the analysis. Two-thirds of novice respondents (61%) were under 20 years of age and 32% reported an age between 20–29 years, only 8% reported ages above 29 years. All novices reported that they are students. Experts had a wide age distribution; 13% of respondents reported being between ages 20–29, 32% between ages 30–39, 22% between ages 40–49, 18% between ages 50–59 years, 11% between ages 60–60 and 4% over 70 years of age. The majority of the experts (82.5%) described themselves as researchers, scientists, lecturers or science logistic professionals, only 17.5% indicated other professions such as teachers, recreation industry professionals, lawyers or charity managers.

## Analysis

**Thematic coding of open-ended responses and words.** Responses to the first question prompt were coded by one researcher to identify the initial broad themes using thematic analysis [54]. The themes were refined by a single researcher, then applied by a team of three researchers for inter-rater reliability measures. Each researcher coded a sample of survey responses, and further refinements of the theme definitions were made. The researchers coded another subset of survey responses and calculated inter-rater reliability at 94% agreement for novices and 95% for expert responses. Disagreement was resolved through discussion amongst coders before the full set of survey responses was coded for the presence and frequency of themes. The final set of themes and their application are detailed in the code book (see S2 File). Words listed in response to the second prompt were coded into eight different themes using the place meaning coding scheme that [30] developed for an Australian tourism destination. We used the five factors that emerged from his factor analysis on visitors' descriptions of the place: i) Natural Attributes, ii) Aesthetic Attributes, iii) Remote/Pristine Character, iv) Cultural

**Table 1. Summary of word analysis separated by the two respondent groups (experts, novices) with descriptive statistics around the number of words listed and their emotional values (valence, arousal, dominance = VAD), and a description of the modifications conducted during data cleaning.** Sentiment values are available for about 78% of words with a VAD value.

| | Valid responses | % Empty | Total # / Unique #, Average # of words | Words w/ VAD value (% of all words listed) | Word modified |
|---|---|---|---|---|---|
| Novice | 142 responses | 4 responses (3%) | 409 words (total) 135 words (unique) (2.9 total words/ respondent) | 385 words (94% have VAD value) | Plural to Singular 16 words (4%), Typo 5 words (1%), Conjugation 6 words (2%), Fragments from phrases/ sentences 109 words (28%) |
| Expert | 93 responses | 2 responses (2%) | 380 words (total) 183 words (unique) (4.1 total words/ respondent) | 362 words (95% have VAD value) | Plural to Singular 17 words (5%), Typo 4 words (1%), Conjugation 7 words (2%), Fragments from phrases/ sentences 65 words (18%) |

Attributes and Values, v) Human Impact as our themes, and we added three more emergent themes (Scientific Attributes; Experience/Adventure/Tourism; Unclear) to cover the range of words that were listed by the respondents in this study. Three coders coded each word individually, and any discrepancies between the coders were resolved through discussion.

**Affective rating of individual words.** To construct the emotional meaning of place, we drew on established emotional ratings for words [44,43,55,56]. The three most important emotional dimensions of words that are described in the theory of emotion [57] are i) valence or pleasantness of the emotions that are evoked by the word and can be described with a range from unhappy to happy; ii) the degree of arousal that is evoked by the word with a range from excited to calm and iii) the dominance or power of the word, which describes the extent to which the word denotes something that ranges from weak-submissive-controlled to strong-dominant-in-control [44,43,55]. Decimal values between 0 and 1 describe the degree of the affective rating. For valence and dominance the value "1" is most positive and most dominant and "0" is most negative and least dominant, and these two affective ratings have a linear relationship (*"words that make people feel happier also making them feel more in control"*, [55, p. 1196]*).* However, arousal has a U-shaped correlation to valence and dominance. More positive/dominant words or negative/weak words are more arousing than neutral words [55]. Emotional stimulation of words is also described through positive and negative polarity [56,58,59]. Positive polarity in a word manifests if a word evokes favorable sentiments and negative polarity is measured if a word is associated with unfavorable sentiments [56]. The polarity is treated as binary using "1" for presence and "0" for absence. In our study, we used the lexicon of over 20,000 English words by [43] to determine ratings for valence, arousal and dominance, and the emotion lexicon by [56,59] with more than 14,000 English words to determine the polarity of each word. We generated the list of words provided by novices and experts and conducted minimal data cleaning following [55] (see Table 1). In the data cleaning process, we corrected all typos (icey = icy), used the emotional values for any inflected forms of words that only had the base form of the word in the dictionary (e.g., sparsely = sparse; glaciers = glacier) and if phrases were listed, we used the meaningful fragments of the phrase (e.g., "large island" became "large" and "island").

We conducted descriptive statics and independent samples t-tests to compare expert and novices' valence, arousal, and dominance measures for the words they reported, as well as the mean percentages of positive emotion words and negative emotion words that experts and novices reported (Table 1).

## Limitations

One limitation of this study lies in the characteristics of the novice population, who were recruited from undergraduate Geoscience classes. While none of the respondents had traveled

to Greenland before, the views and knowledge of Greenland among geoscience students likely differ from that of the general population, Earth Science undergraduate students' views may be more expert-like than those of non-geoscientists. We did not collect demographic data beyond age and profession, and are therefore unable to report ethnic and racial backgrounds of the respondents. Respondents with a racial or ethnic background that is grounded in strong place-meaning (e.g., Indigenous or Native), may show a stronger place meaning or attachment. We focused our recruitment on the Greenland research community. Our results may look differ-ent, had we recruited only Greenlandic residents, likely showing an even stronger place attachment.

## Results

### Thematic Analysis of Open-ended Question: "In no more than two sentences, how do you personally feel about Greenland?"

Most survey respondents in both the novice (65%) and expert (87%) groups had an overall positive impression of Greenland. About a third of novices (31%) had neutral responses (nei-ther positive nor negative), often related to having little to no awareness level of Greenland in their survey response (e.g., "*I have never been to Greenland*"). A small percentage of respon-dents described contradicting feelings about Greenland, with the expert group responses showing a higher percentage of contradictory feelings (8%). Responses were coded as contra-dictory if they noted both positive and negative feelings toward Greenland.

In general, both experts and novices noted observations associated with the environment in terms of its beauty (58% and 29% respectively) and remoteness (37% and 20% respectively), however, nearly half of the experts (49%) also noted observations related to Greenland's people and culture; a theme that was not seen in many of the novice responses (<10%). Fewer novices and experts noted concern for, the importance of, or interest in aspects of Greenland in their responses ($\leq$ 20%). Experts noted emotional connection/attachment to Greenland in the con-text of its environment (56%) and culture (38%) compared to only a few similar responses from novices ($\leq$5%). Novices had either no or only basic awareness of Greenland, while all experts had either developing or complex awareness (see Table 2 for details and SI1 for description of the codes and example responses).

The word clouds in Fig 2 visualize the difference in responses from experts and novices by highlighting the words they used most frequently in larger and bolder font. The findings cor-roborate the thematic coding of the responses; novices use more descriptive words (e.g., place, travel, cool, country, Iceland) and words that show uncertainty (e.g., seem, think), while experts in addition to descriptive terms (e.g., ice, landscape, place) use more words associated with emotions (e.g., feel, love, people), words that illustrate a positive emotional connection (e.g., beautiful, amazing, fascinating) and terms that describe the people of Greenland (e.g., people, culture).

### Descriptive analysis of words ("What words would you use to describe Greenland?")

**Descriptive analysis.** In response to the question "*What words would you use to describe Greenland*?", respondents listed 257 different words. Novices (n = 142) listed a total of 409 words (2.9 words/respondent) of which 35% were unique words. The top words were (Fig 3): *Cold* (68 mentions), *Icy* (45 mentions), *Beautiful* (23 mentions), *Ice* (20 mentions), *Island* (11 mentions), *Large* (10 mentions) and *Big* (9 mentions). Experts (n = 93) listed a total of 380 words (4.1 words/respondent), of these 50% were unique words. The top words were: *Beautiful*

**Table 2. Findings from thematic content analysis and coding of the open-ended responses to the question "*In no more than two sentences, how do you personally feel about Greenland?*" for novice and expert respondents.**

| Code | Frequency | | % of Responses | |
|---|---|---|---|---|
| | Novice | Expert | Novice | Expert |
| **Code 1 Overall Impressions** | | | | |
| Positive | 87 | 84 | 64.9% | 86.6% |
| Negative | 2 | 1 | 1.5% | 1.0% |
| Neutral | 41 | 4 | 30.6% | 4.1% |
| Contradicting | 2 | 8 | 1.5% | 8.2% |
| **Code 2 Environmental Observations/Associations** | | | | |
| Beautiful/Magical/Special Natural Environment | 38 | 56 | 28.4% | 57.7% |
| Remote/Isolated/Challenging/Wild/Ice Covered Natural Environment | 27 | 36 | 20.1% | 37.1% |
| Observations surrounding Greenland's People/Systems/Built Environment | 11 | 47 | 8.2% | 48.5% |
| **Code 3 Concern for Greenland** | | | | |
| Environmental/Scientific Concern | 12 | 20 | 9.0% | 20.6% |
| Cultural/Social/Political Concern | 1 | 12 | 0.7% | 12.4% |
| **Code 4 Greenland's Importance** | | | | |
| Environment/Scientific Importance | 14 | 12 | 10.4% | 12.4% |
| Cultural/Social/Political Importance | 2 | 9 | 1.5% | 9.3% |
| **Code 5 Interest in Greenland** | | | | |
| Environment/Scientific Interest | 14 | 16 | 10.4% | 16.5% |
| Cultural/Social/Political Interest | 3 | 14 | 2.2% | 14.4% |
| **Code 6 Feelings of Connection or Attachment to Greenland** | | | | |
| Environment/Scientific Connection | 7 | 54 | 5.2% | 55.7% |
| Cultural/Social/Political Connection | 2 | 37 | 1.5% | 38.1% |
| **Code 7 Greenland Awareness Level** | | | | |
| No Awareness | 41 | 0 | 30.6% | 0.0% |
| Basic | 73 | 0 | 54.5% | 0.0% |
| Developing | 19 | 15 | 14.2% | 15.5% |
| Complex | 1 | 82 | 0.7% | 84.5% |

(38 mentions), *Vast* (15 mentions), *Wild* (13 mentions), *Remote* (12 mentions), *Nature* (8 mentions), *Changing* (8 mentions) and *Fascinating* (7 mentions). Experts use more words that describe emotions (e.g., beautiful, fascinating, changing), while novices use more descriptive words (e.g., cold, icy, island, northern).

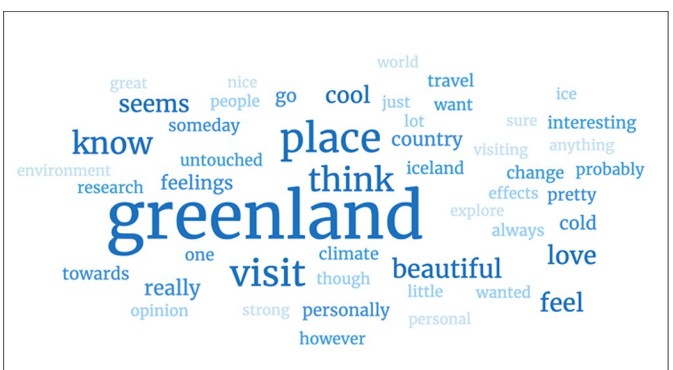 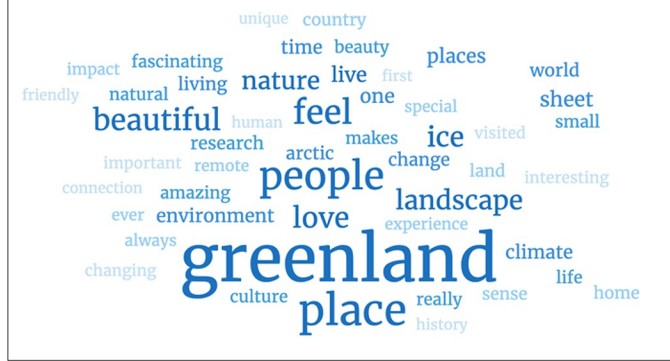

**Fig 2. Word clouds of the novice and expert responses to the question: "*In no more than two sentences, how do you personally feel about Greenland?*" (word cloud left: Novice, right: Expert).**

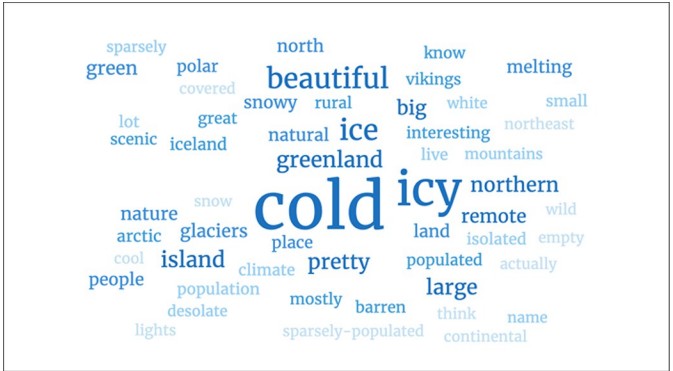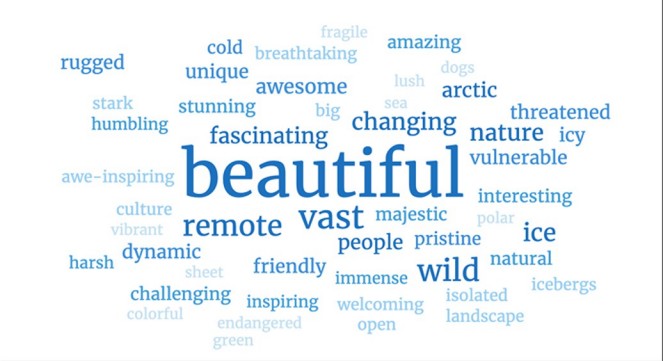

**Fig 3. Word clouds of the novice and expert responses to the question, "W***hat words would you use to describe Greenland***?" (word cloud left: Novice, right: Expert).**

**Emotional value attribution.** Emotional value coding of words used by experts and novices showed measurable differences in the emotional values of the words that novices associated with Greenland when compared to experts (Table 3). Experts associated Greenland with words that hold greater emotional meaning: words used by experts had significantly higher ratings for valence, dominance, and arousal (p < .001; Table 3, Fig 4). Table 4 includes examples of the highest values for valence, arousal, and dominance for both novices and experts. The overall percentage of words of positive polarity was significantly higher in our expert respondents (16% difference to novices) and words of negative polarity was lower (9% difference to novices) with a p < .001 (Table 3). Large effect sizes were found for dominance (d = 1.4) and arousal (d = .86). Medium effect sizes were found for valence (d = .56) and percentage of positive emotion words (d = .62). Negative emotion words had a small to medium effect size (d = .39).

## Categorization of the words

Categorizing the words from both experts and novices by the place meaning themes that [30] developed, showed that two-thirds of novices used natural attributes or descriptions of Greenland (e.g., cold, icy, island), about a quarter of words described aesthetic attributes or the remote or pristine character of Greenland and only 7% of the words described cultural attributes of Greenland (Table 5). About half the experts used aesthetic attributes or highlighted

**Table 3. Descriptive statistics and independent t-test results comparing emotional value ratings of words reported by experts and novices.**

| | | n | Mean (SD) | Median | t | df | p | Cohen's d |
|---|---|---|---|---|---|---|---|---|
| **Valence** | Novice | 92 | 0.59 (0.15) | 0.60 | 4.15 | 228 | < .001 | 0.56 |
| | Expert | 138 | 0.67 (0.13) | 0.69 | | | | |
| **Arousal** | Novice | 92 | 0.45 (0.10) | 0.44 | 6.15 | 166 | < .001 | 0.86 |
| | Expert | 138 | 0.54 (0.13) | 0.54 | | | | |
| **Dominance** | Novice | 92 | 0.46 (0.12) | 0.46 | 10.39 | 218 | < .001 | 1.35 |
| | Expert | 138 | 0.61 (0.10) | 0.61 | | | | |
| | | n | Mean (SD) | Positive/Negative Words (% of words) | t | df | p | Cohen's d |
| **% Positive Emotion Words** | Novice | 92 | 17.24 (26.42) | 75 (17%) | 4.61 | 228 | < .001 | 0.62 |
| | Expert | 138 | 33.4 (25.43) | 113 (33%) | | | | |
| **% Negative Emotion Words** | Novice | 92 | 24.88 (27.67) | 88 (25%) | -3.21 | 228 | 0.001 | 0.39 |
| | Expert | 138 | 15.58 (17.58) | 63 (16%) | | | | |

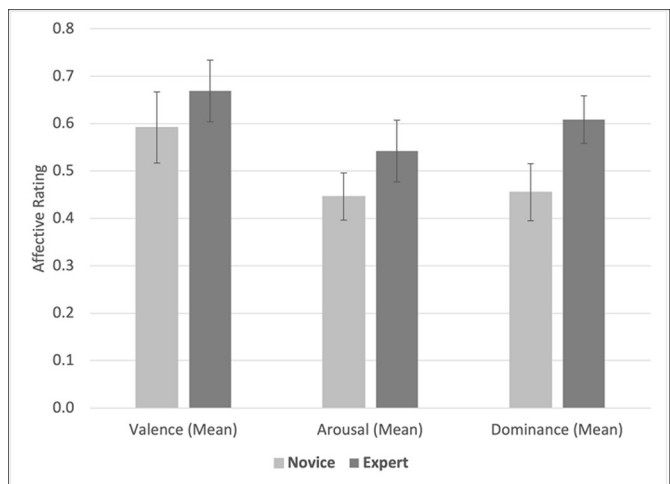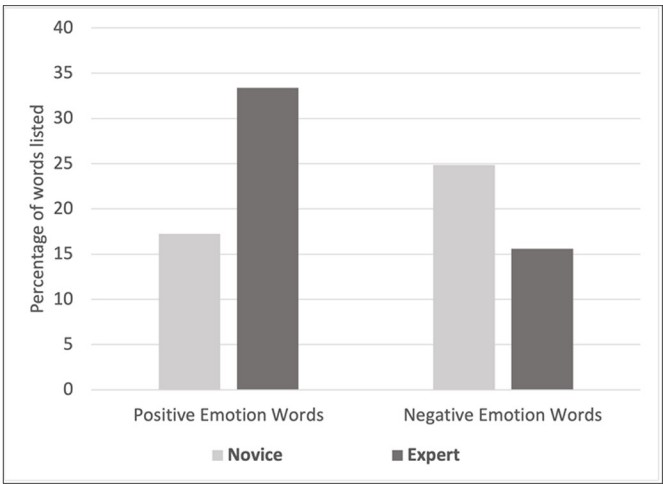

**Fig 4. Results of emotional value coding of the words differentiated by novices and experts.** Differences are all statistically significant. *Left*: Emotional value analysis shows the mean values and standard deviation. *Right*: Percentage of positive and negative polarity of words.

Greenland's remote or pristine character and about 11% of the experts included references to cultural attributes of Greenland. A small number of the experts (5%) described the human impact on Greenland, however, even fewer novices' (0.5%) words described these impacts.

## Discussion

### A new approach to measuring sense of place in novices and experts

Sense of place is being measured through a variety of quantitative and qualitative instruments and approaches (review in [4]) but is usually studied in cases where participants have in-person place experiences and is based on self-reported data. Here we introduce a novel approach that combines tools from cognitive psychology and linguistics with established instruments to quantify the emotional value of words, categorize them individually, and code the meaning of short answer responses used by survey respondents to describe Greenland–a place that only the expert-like respondents have visited in person. These approaches build on the understanding that words carry important meaning and emotion [43], and we use them here as a proxy

**Table 4. List of the highest and lowest value words for valence, arousal and dominance.** Note that arousal has a u-shaped distribution, while valence and dominance show a linear correlation from positive (high) to negative (low).

| | Novice | | | Expert | | |
|---|---|---|---|---|---|---|
| | Word (Valence) | Word (Arousal) | Word (Dominance) | Word (Valence) | Word (Arousal) | Word (Dominance) |
| High (1st) | Magnificent (1) | Threatened (0.928) | Breathtaking (0.921) | Perfect (0.98) | Threatened (0.928) | Power (0.953) |
| High (2nd) | Love (1) | Wild (0.922) | Awesome (0.891) | Awesome (0.98) | Wild (0.922) | Majority (0.93) |
| High (3rd) | Awesome (0.98) | Awesome (0.897) | Extremely (0.886) | Fascinating (0.979) | Awesome (0.897) | Breathtaking (0.9211) |
| High (4th) | Fascinating (0.979) | Endangered (0.885) | Magnificent (0.886) | Wonderful (0.971) | Endangered (0.885) | Majestic (0.906) |
| High (5th) | Wonderful (0.971) | Adventure (0.857) | Political (0.875) | Freedom (0.969) | Amazing (0.837) | Force (0.905) |
| Low (1st) | Threatened (0.052) | Calm (0.1) | Empty (0.188) | Death (0.031) | Tranquil (0.094) | Empty (0.188) |
| Low (2nd) | Endangered (0.083) | Peaceful (0.108) | Small (0.12) | Threatened (0.052) | Calm (0.1) | Fragile (0.163) |
| Low (3rd) | Desolate (0.104) | Natural (0.118) | Unstable (0.15) | Endangered (0.083) | Quiet (0.105) | Sad (0.149) |
| Low (4th) | Raw (0.143) | Water (0.123) | Unknown (0.204) | Desolate (0.104) | Calming (0.105) | Neglected (0.16) |
| Low (5th) | Unstable (0.167) | Serene (0.132) | Sparse (0.209) | Racism (0.122) | Peaceful (0.108) | Mosquito (0.198) |

**Table 5. Percentage of words used to describe Greenland relative to the total number of words used by experts and novices across themes that emerged from the place meaning study by [30].**

| | Novice | Novice Examples (most frequently used words) | Experts | Expert Examples (most frequently used words) |
|---|---|---|---|---|
| **Natural Attributes and Description** | 59.7% | cold, icy, ice, island, large | 24.3% | vast, nature, Arctic, ice, icy, rugged |
| **Aesthetic Attributes, Descriptions and Feelings** | 15.8% | beautiful, pretty, interesting, unique, cool | 36.6% | beautiful, fascinating, awesome, unique, amazing |
| **Remote or Pristine Character** | 8.2% | remote, sparse, untouched, wild, pristine | 11.6% | wild, remote, pristine, harsh, isolated |
| **Cultural Attributes and Values** | 7.0% | people, population, Viking, Denmark, native | 10.9% | friendly, people, culture, welcoming, home |
| **Human Impact on Greenland** | 0.5% | threatened, endangered | 5.4% | changing, threatened, vulnerable, endangered, infrastructure |
| **Experience/Adventure/Tourism** | 1.4% | exotic, adventure, fishing, hiking, tourist | 1.0% | exotic, fishing, hunting, kayaking |
| **Scientific/Research Attributes** | 4.3% | melting, research, biodiversity, climate, ecological | 1.2% | research, hydrology, science, elemental |
| **Unclear** | 3.4% | rural, green, clean, Greenland, place | 9.4% | dynamic, stark, complex, complicated, force |

for someone's sense of place. Each of the three approaches (emotional coding, categorization, thematic coding) independently reveals significant differences between the way novices and experts describe Greenland and therefore can be used to quantify where a person falls on the novice to expert continuum (Fig 1). This is the first approach that does not rely on self-report data but instead quantifies proxy data to infer an emotional connection to a place. While the methodological approach was developed for Greenland, we believe that the codes and categories can be applied to other polar and alpine places.

## In which ways do novices' sense of place differ from experts' sense of place?

We found both a more nuanced place meaning of Greenland by experts when compared to novices, as well as expressions of connection, awareness, and feeling toward Greenland. The lack of awareness of, or sentiments towards, Greenland is an important characteristic of the novices in this study. Experts show a much higher cultural, historic and social awareness of Greenland than novices. Some experts describe contradicting feelings about Greenland, which were often grounded in a love for the natural beauty of the island paired with an awareness of the complex cultural history and environmental or social challenges. The difference between novice and expert data cuts across multiple dimensions (Tables 2 and 5), with the strongest difference between expert and novice responses playing out in connections to, concerns for, and importance of Greenland. The emotional value coding allowed an exploration of the emotions that novices and experts connect with Greenland. Experts frequently used words that carry emotions. In contrast, novices used descriptive words that illustrated a superficial knowledge of Greenland (Tables 3–5). Feelings of connection or emotional attachment to a place are important components of sense of place [53,60]. Following the definition that sense of place includes both an awareness of the physical location as well as an emotional and cultural component [4], experts' focus on social, cultural and political aspects of Greenland is likely a reflection of a strong place attachment, a feeling that is mostly absent in novices.

While we see significant differences in the types of attributes used to describe Greenland and the awareness and concern around cultural and social topics, we found that the environmental or scientific importance of Greenland is expressed at similar levels by experts and novices, possibly a representation that much of the communication about Greenland focuses on its unique role in the global climate system [61]. While the respondents that had not yet traveled to Greenland before (novices) built their knowledge about Greenland from sources of information (e.g., news, social media, books, magazine articles, films) and not personal

experiences, those who had traveled to or lived in Greenland (experts) grounded their views and feelings about Greenland in direct experiences and an immersion in the culture. If geoscience education aims to increase learners' sense of place, it is important to expose learners to materials that build awareness of the culture, history and peoples of Greenland in addition to the scientific and environmental concepts.

## How can we explain the difference in the view of Greenland between novices and experts?

Sense of place is usually formed through personal experiences [26], however, sense of place around far-away, remote locations is built through exposure to information about the place. This information can be part of intentional branding for a place or random information. Intentional place branding for destinations, regions or nations uses stories or narratives about a location to differentiate one place from another and uses cultural images and representations of place [61,62]. Place branding according to [63] includes the development of a "holistic narrative" around locally relevant and unique elements of a place considered of value to the intended marketing. Branding efforts that are conducted by different organizations often result in incompatible or incoherent narratives, building multi-faceted views of places [64,65]. The traditional and most-wide spread place branding efforts around Greenland are designed as tourism information; documentaries and magazines usually highlight Greenland as a place of nature. In these traditional branding efforts, Greenland is coined as a "*colossal, remote and frozen landscape in silence and solitude. . . a frozen Paradise on Earth, inhabited by blissful indigenous people living harmoniously with, in and even 'as' nature*" [61, p. 445] or alternatively narratives focus on adventure, exploration and discovery. More recent branding efforts have been initiated on the backdrop of increased attention on natural resources, geopolitical changes, changing climate and local efforts for political and economic independence. These recent branding efforts highlight sustainability efforts, modern architecture and mobilization [61,65]), while the reality of travel to and life in Greenland includes climate-impacted changes in the landscape, mineral extraction industry, shortage of qualified workers, transportation challenges and complexity for Greenlandic people [61].

While novices in our study have an overall positive view of Greenland, they display low awareness levels and little concern, interest or knowledge of the culture and history of the island and hold a view of Greenland that is not nuanced and does not include much of the more recent branding content. Novices' impressions of Greenland focus on the natural environment and hold low emotional value. Experts on the other hand display a more differentiated view and picture of Greenland. While it may be an expected and intuitive finding that those who experience a place directly have a more differentiated view, it is striking that novices in our study strongly subscribe to the traditional place branding that exists around Greenland, a phenomenon that [30] also described for Australia. Efforts in place branding that focus on an expanded view of Greenland and the importance of the culture or thoughtful geoscience instruction, may be able to expand novices' views.

## Why is fostering a sense of place important, especially for a far-away place?

Studies of place-based science education have shown that connecting instructional content to places that are relevant to the learner improves their learning outcomes [5,25,49]. Place-based education helps students develop more nuanced place meanings which can influence their cognitive outcomes (e.g., factual, conceptual and procedural knowledge), behavioral outcomes (e.g., pro-environmental behaviors) and affective outcomes (e.g., increased interest in the discipline, increased connection to the environment) [4]. Place-based education draws from

socio-cultural theories of learning to frame the development of curricula; theories that emphasize learning as a social process infusing elements of language and culture [66] and learning done through authentic activity [67,68]. While most place-based education builds on and strengthens direct personal connections to a place, we propose that in a globalized world with readily available access to immersive and virtual imagery and tools, place-based education does not need to be limited to tangible places that learners have experienced. Future research is needed to explore whether virtual field experiences [27,28,34–37] and augmented reality [69,70]

make it possible for learners to 'travel' to far-away places and use immersive tools for exploration. Educational research has shown the efficacy of immersive experiences and virtual field trips on student learning [27] and others have identified additional outcomes achieved by immersive virtual experiences including increased inclusion and access as compared to in-person field experiences [71,72] and increased engagement and interest to see a place one's never been to [27]. Virtual field experiences can provide the opportunity to engage in authentic activities, such as environments designed for teaching geologic field mapping (e.g., [73]). The concept of "presence" can help frame the development of virtual field experiences in which the student feels a sense of "being there" in the natural environment and remembers their experience in the virtual environment as having visited a "place" rather than just viewing images on a computer [74]. While socio-cultural elements of places may be more difficult to represent virtually, there have been efforts to incorporate the cultural elements of a place into virtual environments [75]. For these reasons, some argue that virtual environments can be intentionally designed to foster a student's sense of place [76]. However, this has yet to be a focus of virtual or immersive experiences. Given what we know about the efficacy of local place-based education, we suggest that learners who develop more advanced place meaning will likely see improved learning outcomes as they virtually travel to a far-away place and learn about local phenomena. The next steps in our research will be to study if place-based virtual experiences can lead to deeper place meaning for far-away places and, in turn, improved student outcomes.

## Conclusion

Our new methodological approach quantifies sense of place using the emotional value of words and phrases that respondents used to describe their perception of a place as a proxy. Positive learning outcomes for place-based education rely on a strong sense of place, thus the characterization of learners' sense of place can effectively guide instruction. We found a statistically significant difference between novices' and experts' sense of place as expressed in their responses. Following established approaches from cognitive psychology, individual words that respondents shared to describe their association with Greenland allowed us to quantify the emotional value of these words, and we used thematic coding to further categorize the findings. Words and expressions used in experts' survey responses showed stronger emotional values, awareness of the culture and people, feelings of connection, and concern about and interest for the place than novices'. The findings indicate that this new approach to characterizing sense of place is valid and robust. We suggest that fostering broader perspectives and complex place meaning in learners will increase their sense of place. We suggest that future research is needed to study if virtual visits to remote far-away places can result in an increased sense of place in learners. An increased sense of place may result in positive learning outcomes around geoscience topics and possibly even increase the learners' hope and desire for environmental stewardship around the place.

## Supporting information

**S1 File.**
(XLSX)

**S2 File.**
(DOCX)

## Acknowledgments

We are grateful for contributions from Lynne Harden on the research design and discussions with Steven Semken on the framing. Meghan Henderson wrote the Excel Macros that allowed an easy lookup of words in the lexica. Ami Nacu-Schmidt created the illustration in Fig 1.

## Author Contributions

**Conceptualization:** Anne U. Gold, Emily M. Geraghty Ward, Megan K. Littrell.

**Data curation:** Anne U. Gold, Emily M. Geraghty Ward, Casey L. Marsh.

**Formal analysis:** Anne U. Gold, Emily M. Geraghty Ward, Megan K. Littrell.

**Funding acquisition:** Anne U. Gold, Twila A. Moon, Spruce W. Schoeneman, Alia L. Khan.

**Investigation:** Anne U. Gold.

**Methodology:** Anne U. Gold, Emily M. Geraghty Ward, Megan K. Littrell.

**Project administration:** Anne U. Gold, Twila A. Moon, Spruce W. Schoeneman, Alia L. Khan.

**Validation:** Anne U. Gold, Emily M. Geraghty Ward, Casey L. Marsh.

**Visualization:** Anne U. Gold, Emily M. Geraghty Ward, Casey L. Marsh.

**Writing – original draft:** Anne U. Gold, Emily M. Geraghty Ward.

**Writing – review & editing:** Anne U. Gold, Emily M. Geraghty Ward, Casey L. Marsh, Twila A. Moon, Spruce W. Schoeneman, Alia L. Khan, Megan K. Littrell.

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
