## [Decision Letter · Decision Letter 0]

12 Dec 2022

PONE-D-22-27793Measuring novice-expert sense of place for a far-away place: Implications for geoscience instruction

PLOS ONE

Dear Dr. Gold,

Thank you for submitting your manuscript to PLOS ONE. After careful consideration, we feel that it has merit but does not fully meet PLOS ONE’s publication criteria as it currently stands. Therefore, we invite you to submit a revised version of the manuscript that addresses the points raised during the review process.

Please note that we have only been able to secure a single reviewer to assess your manuscript. We are issuing a decision on your manuscript at this point to prevent further delays in the evaluation of your manuscript. Please be aware that the editor who handles your revised manuscript might find it necessary to invite additional reviewers to assess this work once the revised manuscript is submitted. However, we will aim to proceed on the basis of this single review if possible. 

We look forward to receiving your revised manuscript.

Kind regards,

Alice Coles-Aldridge

Staff Editor

PLOS ONE

Journal Requirements:

a) Did participants provide their written or verbal informed consent to participate in this study?

   "The research was funded as part of the PolarPASS project by the National Science Foundation under awards OPP-2021503, OPP-2021275 and OPP-2021543 and National Oceanic and Atmospheric Administration through award NA17OAR4320101. We are grateful for contributions from Lynne Harden on the research design and discussions with Steven Semken on the framing. Meghan Henderson wrote the Excel Macros that allowed an easy lookup of words in the lexica. Ami Nacu-Schmidt created the illustration in figure 1. We received IRB approval for this study from the University of Colorado IRB office under protocol 21-0189."

 "OPP-2021503

OPP-2021275

OPP-2021543"

6. Please ensure that you include a title page within your main document. You should list all authors and all affiliations as per our author instructions and clearly indicate the corresponding author.

7. Your ethics statement should only appear in the Methods section of your manuscript. If your ethics statement is written in any section besides the Methods, please move it to the Methods section and delete it from any other section. Please ensure that your ethics statement is included in your manuscript, as the ethics statement entered into the online submission form will not be published alongside your manuscript. 

Reviewers' comments:

Reviewer's Responses to Questions

**Comments to the Author**

1. Is the manuscript technically sound, and do the data support the conclusions?

Reviewer #1: Yes

2. Has the statistical analysis been performed appropriately and rigorously? 

Reviewer #1: Yes

3. Have the authors made all data underlying the findings in their manuscript fully available?

Reviewer #1: Yes

4. Is the manuscript presented in an intelligible fashion and written in standard English?

Reviewer #1: Yes

5. Review Comments to the Author

Reviewer #1: I thought this was a very well-organized, readable manuscript. I learned a great deal about the methodology of using this approach to analyzing sense of place. The only questions I had about the manuscript were the limitations imposed by the two groups--novice and expert--and how these groups were defined. The novice group was mostly young, from a research university. No racial or ethnic demographics were provided. The expert group was highly-educated, older, more experienced. So I wondered first if the novice group had been made up of faculty and other older professionals would they have responded more closely to the expert group? Undergraduates might have a less nuanced understanding of the cultural and geopolitical aspects of a place they've never been than university faculty would, regardless of whether they had been there. Likewise, if the novice group had included students from indigenous populations in the U.S., they might have a very different way of responding than the U Colorado Boulder students.

I thought the limitation of two sentences for the first question in your survey might actually support a less nuanced picture. What would have changed in your research if you had left the amount they wrote open-ended? Some people think while they write so they might reach some of the more complex issues if they were expected to write more. They might reflect more on what they have heard or know about a place rather than hitting on their first impression (cold, remote, beautiful, melting).

Another thought I had was that building knowledge or culture and other aspects of a place is important, but how will you distinguish in your follow-up research whether it's the virtual experience or the type of content that is the most significant in understanding place. For example, I could see where incorporating videos of various people who live in Greenland into your virtual field trip might influence students' perception of that place. Adding readings (novels or articles) that highlight this content might be equally useful. I think you will need to be very careful in distinguishing whether it's the media or the message that matters. I have been very impacted by just meeting one other person from a place I have never been, which can be very influential to how you understand a place. Incorporating a virtual talk from someone who lives there, possibly from the indigenous culture or a faculty member, could be very influential on what someone knows about a place and how they feel about it.

Overall I thought this was extremely well-written and very thought provoking. Thanks so much for providing me the opportunity to review.

6. PLOS authors have the option to publish the peer review history of their article (what does this mean?). If published, this will include your full peer review and any attached files.

Reviewer #1: **Yes: **Diana M Dalbotten

---

## [Author Response · Author response to Decision Letter 0]

23 Jan 2023

[copied from response to reviewer file]

Response to Editor and Reviewer Comments and Requests

Editor Request: Please ensure that your manuscript meets PLOS ONE's style requirements, including those for file naming. 

How we addressed it: We updated the file formatting, and it now meets the PLOS One style requirements. 

Editor Request: Please amend your current ethics statement to address the following concerns:

a) Did participants provide their written or verbal informed consent to participate in this study?

How we addressed it: In our study, we collected electronic consent from all survey respondents. We added the following sentence to the ethics statement (located now within Methodology, subsection Survey Instrument: All participants provided electronic consent to participate in the study.

Editor Request: We note that the grant information you provided in the ‘Funding Information’ and ‘Financial Disclosure’ sections do not match. When you resubmit, please ensure that you provide the correct grant numbers for the awards you received for your study in the ‘Funding Information’ section.

Thank you for stating the following in the Acknowledgments Section of your manuscript: 

 "The research was funded as part of the PolarPASS project by the National Science Foundation under awards OPP-2021503, OPP-2021275 and OPP-2021543 and National Oceanic and Atmospheric Administration through award NA17OAR4320101. We are grateful for contributions from Lynne Harden on the research design and discussions with Steven Semken on the framing. Meghan Henderson wrote the Excel Macros that allowed an easy lookup of words in the lexica. Ami Nacu-Schmidt created the illustration in figure 1. We received IRB approval for this study from the University of Colorado IRB office under protocol 21-0189."

 "OPP-2021503

OPP-2021275

OPP-2021543"

How to address – our response:

We removed the funding information from the Acknowledgement section. The funding information and role of the funder is now added to the coverletter. It should read as follows: 

“National Science Foundation awards OPP-2021503, OPP-2021275 and OPP-2021543 

National Oceanic and Atmospheric Administration award NA17OAR4320101. The funders had no role in study design, data collection and analysis, decision to publish, or preparation of the manuscript.”

Editor Request: In your Data Availability statement, you have not specified where the minimal data set underlying the results described in your manuscript can be found. PLOS defines a study's minimal data set as the underlying data used to reach the conclusions drawn in the manuscript and any additional data required to replicate the reported study findings in their entirety. All PLOS journals require that the minimal data set be made fully available. For more information about our data policy, please see http://journals.plos.org/plosone/s/data-availability.

How we addressed it: We have uploaded a file that includes all the data to replicate the study. All data is anonymous and not identifiable. 

Editor Request: Please ensure that you include a title page within your main document. You should list all authors and all affiliations as per our author instructions and clearly indicate the corresponding author.

How we addressed it: We have reformatted the title page to match the PLOS One guidelines: Title page: List title, authors, and affiliations as the first page of the manuscript. We have now clearly indicated the corresponding author. Note that our long title does not exceed the character limit of the short title. We, therefore, suggest using the title both as the long and the short title. If you prefer to have a short title that is shorter than the long title please use as the short title: “Measuring novice-expert sense of place for a far-away place”. 

Editor Request: Your ethics statement should only appear in the Methods section of your manuscript. If your ethics statement is written in any section besides the Methods, please move it to the Methods section and delete it from any other section. Please ensure that your ethics statement is included in your manuscript, as the ethics statement entered into the online submission form will not be published alongside your manuscript. 

How we addressed it: We moved the ethics statement to the sub-section “Survey instrument” in the “Methodology” section

Reviewers' comments:

Reviewer's Responses to Questions

Comments to the Author

1. Is the manuscript technically sound, and do the data support the conclusions?

Reviewer #1: Yes

2. Has the statistical analysis been performed appropriately and rigorously? 

Reviewer #1: Yes

3. Have the authors made all data underlying the findings in their manuscript fully available?

Reviewer #1: Yes

4. Is the manuscript presented in an intelligible fashion and written in standard English?

Reviewer #1: Yes

5. Review Comments to the Author

Reviewer #1: I thought this was a very well-organized, readable manuscript. I learned a great deal about the methodology of using this approach to analyzing sense of place. The only questions I had about the manuscript were the limitations imposed by the two groups--novice and expert--and how these groups were defined. The novice group was mostly young, from a research university. No racial or ethnic demographics were provided. The expert group was highly-educated, older, more experienced. So I wondered first if the novice group had been made up of faculty and other older professionals would they have responded more closely to the expert group? Undergraduates might have a less nuanced understanding of the cultural and geopolitical aspects of a place they've never been than university faculty would, regardless of whether they had been there. Likewise, if the novice group had included students from indigenous populations in the U.S., they might have a very different way of responding than the U Colorado Boulder students.

I thought the limitation of two sentences for the first question in your survey might actually support a less nuanced picture. What would have changed in your research if you had left the amount they wrote open-ended? Some people think while they write so they might reach some of the more complex issues if they were expected to write more. They might reflect more on what they have heard or know about a place rather than hitting on their first impression (cold, remote, beautiful, melting).

Another thought I had was that building knowledge or culture and other aspects of a place is important, but how will you distinguish in your follow-up research whether it's the virtual experience or the type of content that is the most significant in understanding place. For example, I could see where incorporating videos of various people who live in Greenland into your virtual field trip might influence students' perception of that place. Adding readings (novels or articles) that highlight this content might be equally useful. I think you will need to be very careful in distinguishing whether it's the media or the message that matters. I have been very impacted by just meeting one other person from a place I have never been, which can be very influential to how you understand a place. Incorporating a virtual talk from someone who lives there, possibly from the indigenous culture or a faculty member, could be very influential on what someone knows about a place and how they feel about it.

Overall I thought this was extremely well-written and very thought provoking. Thanks so much for providing me the opportunity to review.

6. PLOS authors have the option to publish the peer review history of their article (what does this mean?). If published, this will include your full peer review and any attached files.

Do you want your identity to be public for this peer review? For information about this choice, including consent withdrawal, please see our Privacy Policy.

Reviewer #1: Yes: Diana M Dalbotten

How did we address reviewer comments:

1) Limitations of study 

Your point is well-taken. The make-up of the pool of respondents is of course very important to the results. The reason why we chose the undergraduate college student as our novice population is that this is a baseline study as part of a larger study in which we implement immersive experiences in undergraduate classrooms. For the follow-on study, we have recruited undergraduate students from a diversity of schools (including students with Indigenous backgrounds and students with a background that is traditionally underrepresented in STEM disciplines). For this study, we did not ask respondents about their racial or ethnic identity. 

To address this point in the manuscript, we have added the following text to the Limitations section: ”We did not collect demographic data beyond age and profession, and are therefore unable to report ethnic and racial backgrounds of the respondents. Respondents with a racial or ethnic background that is grounded in strong place-meaning (e.g., Indigenous or Native), may show a stronger place meaning or attachment. We focused our recruitment on the Greenland research community. Our results may look different, had we recruited only Greenlandic residents, likely showing an even stronger place attachment.”

In addition, we have added the following test to the Study Participant section: “We selected U.S. undergraduate college students as our novice population because very few students at a U.S. college would have visited Greenland before, thus, responding to the survey with the expertise of a novice. The reason why we surveyed undergraduate students is because this study is part of a larger study that aims to improve undergraduate college instruction using immersive virtual field experiences and therefore, responses from college students provide a meaningful baseline for novices.” [and further on in the same paragraph, we added about the experts] “These networks include many people who are conducting research in Greenland, live in Greenland or spent time in Greenland, and thus provide access to people who have visited, worked, lived or currently live in Greenland.”

2) Changes in the way open-ended question was asked

The reason why we limited the writing to two sentences is that we wanted to capture what first comes to mind for the respondents. We further wanted to keep the survey short and be able to compare the results of different respondents. Had we left the question without guidance on length, one respondent would have written 10 sentences, while others may have written just one sentence. Providing guidance to keep the answers short allowed for responses that were comparable between all respondents. 

To address this point in the manuscript, we have added the following sentence to the Methodology section: “In asking respondents to keep their answer to two sentences, we forced sharing of the initial associations with the place and allowed for comparable response length between study subjects.“ 

3) Follow on research limitation:

This is a very good point and helpful advice. We will keep this in mind for our future study design. 

How we addressed this: We uploaded all four figures to PACE and they look clear. See screenshot below.

---

## [Decision Letter · Decision Letter 1]

11 Jul 2023

PONE-D-22-27793R1Measuring novice-expert sense of place for a far-away place: Implications for geoscience instructionPLOS ONE

Dear Dr. Gold,

Thank you for submitting your manuscript to PLOS ONE. After careful consideration, we feel that it has merit but does not fully meet PLOS ONE’s publication criteria as it currently stands. Therefore, we invite you to submit a revised version of the manuscript that addresses the points raised during the review process.

We look forward to receiving your revised manuscript.

Kind regards,

Kendra Helen Oliver, Ph.D.

Academic Editor

PLOS ONE

Journal Requirements:

Additional Editor Comments:

Hello,

I hope this letter finds you well. I am writing to provide feedback on the manuscript titled "Measuring novice-expert sense of place for a far-away place: Implications for geoscience instruction." I would like to commend you on this work and the significant adjustments you have made in addressing the reviewers' comments. The revisions have certainly improved the clarity and quality of your work. However, there is one key aspect that still requires attention, as highlighted by Reviewer 2.

Reviewer 2 pointed out the need for reframing the hypothesis to clearly elucidate the role of the virtual experience in the context of measuring novice-expert sense of place. This is a valid concern and addressing it will greatly enhance the overall contribution of your paper. To this end, I suggest making the following revisions:

Introduction:

a. Clearly state the objective of your study, emphasizing the investigation of how virtual experiences contribute to measuring novice-expert sense of place for a far-away location.

b. Provide a brief overview of the existing literature on the use of virtual experiences in geoscience instruction and their potential impacts on developing a sense of place.

Hypothesis:

a. Revisit your hypothesis and reframe it to explicitly incorporate (or remove the mention of) the role of the virtual experience. Consider highlighting how the virtual experience affects novice-expert sense of place formation and whether it influences participants' spatial understanding, emotional connection, or both.

b. Justify the need for investigating the impact of the virtual experience, particularly in the context of assessing sense of place for a far-away location where physical visits may be limited.

Results and Discussion:

a. Analyze and interpret the data collected in the context of the reframed hypothesis, emphasizing the relationship between the virtual experience and participants' sense of place.

b. Discuss any limitations or challenges associated with using a virtual experience as a proxy for a physical visit, and address how these limitations may impact the interpretation of the results.

By addressing these points, you will provide a clearer understanding of the role and significance of the virtual experience in your study. This reframing will contribute to the broader discussion surrounding the use of virtual experiences in geoscience instruction and the measurement of sense of place. Once again, I would like to express my appreciation for the efforts you have made in revising your manuscript. I believe that incorporating these suggested revisions will strengthen the impact and relevance of your work. Should you have any questions or require further clarification, please do not hesitate to reach out to me.

Thank you for your attention to these revisions. I look forward to reviewing the updated version of your manuscript.

Kendra H. Oliver, Ph.D., M.P.S.

Reviewers' comments:

Reviewer's Responses to Questions

**Comments to the Author**

1. If the authors have adequately addressed your comments raised in a previous round of review and you feel that this manuscript is now acceptable for publication, you may indicate that here to bypass the “Comments to the Author” section, enter your conflict of interest statement in the “Confidential to Editor” section, and submit your "Accept" recommendation.

Reviewer #1: All comments have been addressed

Reviewer #2: (No Response)

2. Is the manuscript technically sound, and do the data support the conclusions?

Reviewer #1: Yes

Reviewer #2: No

3. Has the statistical analysis been performed appropriately and rigorously? 

Reviewer #1: Yes

Reviewer #2: Yes

4. Have the authors made all data underlying the findings in their manuscript fully available?

Reviewer #1: Yes

Reviewer #2: Yes

5. Is the manuscript presented in an intelligible fashion and written in standard English?

Reviewer #1: Yes

Reviewer #2: Yes

6. Review Comments to the Author

Reviewer #1: This article looks great. Thanks for the chance to review. All of my comments were addressed in full.

Reviewer #2: The manuscript titled "Measuring novice-expert sense of place for a far-away place: Implications for geoscience instruction" by Gold et al. introduces an innovative method to measure individuals' sense of place for remote locations. The objective is to evaluate whether survey data can support the idea that virtual visits to distant locations can enhance learners' sense of place, transitioning them from a novice-like to a more expert-like understanding. The authors discuss virtual learning experiences, behavioral changes, and introduce novel tools for measuring sense of place in remote areas. They argue that virtual experiences can indeed alter learners' sense of place. The study emphasizes the significance of a strong sense of place for positive learning outcomes in place-based education. The authors' approach, quantifying sense of place based on emotional value of words and phrases, can provide guidance for instructional design and assist educators in tailoring lessons to learners' perceptions of a place. This research holds substantial importance and is likely to be of interest to a wide range of professionals in the field.

However, while the methodological approach appears appropriate, I came across a significant issue that restricts the discussion of the results. The hypothesis regarding the deeper meaning of place resulting from virtual experiences was not explored in this study. Although the study reported data on vocabulary differences between novice and expert learners in describing Greenland as a sense of place proxy, it remains ambiguous since no virtual learning experiences were integrated into the results. I would like to clarify that this critique does not pertain to the data quality; I read the manuscript with great interest. However, the data fail to address the proposed question. It becomes challenging to assess the significance of survey data in terms of changing learners' sense of place because there is no integration with virtual learning experiences to support the argument that virtual visits to remote locations can induce behavioral change. Employing surveys both before and after virtual visits to evaluate shifts in awareness would have likely been a more effective strategy when employing this methodology. Undoubtedly, there exists a connection between vocabulary and behavior. Words serve as lenses through which we perceive the world. Language has the potential to influence opinions and behaviors, and vocabulary allows us to interpret and express ourselves. However, the authors need to provide further justification for the suitability of survey data in describing their hypothesis. Although the approach of examining differences between novice and expert use of words and phrases to describe Greenland is intriguing, it should be employed as preliminary data, as it lacks integration with virtual experiences.

7. PLOS authors have the option to publish the peer review history of their article (what does this mean?). If published, this will include your full peer review and any attached files.

Reviewer #1: No

Reviewer #2: No

---

## [Author Response · Author response to Decision Letter 1]

21 Aug 2023

Response to Reviewers

Reviewer and Editor Comments: 

Reviewer 2 pointed out the need for reframing the hypothesis to clearly elucidate the role of the virtual experience in the context of measuring novice-expert sense of place. This is a valid concern and addressing it will greatly enhance the overall contribution of your paper. To this end, I suggest making the following revisions:

Introduction:

a. Clearly state the objective of your study, emphasizing the investigation of how virtual experiences contribute to measuring novice-expert sense of place for a far-away location.

b. Provide a brief overview of the existing literature on the use of virtual experiences in geoscience instruction and their potential impacts on developing a sense of place.

Hypothesis:

a. Revisit your hypothesis and reframe it to explicitly incorporate (or remove the mention of) the role of the virtual experience. Consider highlighting how the virtual experience affects novice-expert sense of place formation and whether it influences participants' spatial understanding, emotional connection, or both.

b. Justify the need for investigating the impact of the virtual experience, particularly in the context of assessing sense of place for a far-away location where physical visits may be limited.

Results and Discussion:

a. Analyze and interpret the data collected in the context of the reframed hypothesis, emphasizing the relationship between the virtual experience and participants' sense of place.

b. Discuss any limitations or challenges associated with using a virtual experience as a proxy for a physical visit, and address how these limitations may impact the interpretation of the results.

How did we address the comment: 

The concern that reviewer 2 raised is valid and we appreciate you laying out how to address the concern. We decided to follow your recommendation and remove the mention of virtual field trips from the hypothesis and framing of the paper and instead we mention that the study may have implications on virtual field trips. All changes are provided in a version that includes “track changes”. Specifically we changed: 

Abstract – removed mention of virtual field trips in the motivation and hypothesis and focused on the research design that compares novices and experts. At the end of the abstract, we rephrased a sentence to the findings may have implications for virtual field trips in education instead of using it as a motivation. 

Introduction – removed the mention of virtual field trips as part of the motivation and instead rephrased that the study may have implications for virtual field trips in education. 

Results: No changes since we are not including data around virtual field trips

Discussion: We rephrased the text to suggest that future research may explore the impacts on virtual field trips and removed any connection to the motivation of the study.

Conclusion: We rephrased the text to remove mention of the virtual field trips as part of the study motivation and instead phrased it to suggest that future research may be needed to study the impact on virtual field trips in education.

---

## [Decision Letter · Decision Letter 2]

4 Oct 2023

Measuring novice-expert sense of place for a far-away place: Implications for geoscience instruction

PONE-D-22-27793R2

Dear Dr. Gold,

We’re pleased to inform you that your manuscript has been judged scientifically suitable for publication and will be formally accepted for publication once it meets all outstanding technical requirements.

Kind regards,

Kendra Helen Oliver, Ph.D.

Academic Editor

PLOS ONE

Additional Editor Comments (optional):

Reviewers' comments:

Reviewer's Responses to Questions

**Comments to the Author**

1. If the authors have adequately addressed your comments raised in a previous round of review and you feel that this manuscript is now acceptable for publication, you may indicate that here to bypass the “Comments to the Author” section, enter your conflict of interest statement in the “Confidential to Editor” section, and submit your "Accept" recommendation.

Reviewer #2: All comments have been addressed

2. Is the manuscript technically sound, and do the data support the conclusions?

Reviewer #2: Yes

3. Has the statistical analysis been performed appropriately and rigorously? 

Reviewer #2: Yes

4. Have the authors made all data underlying the findings in their manuscript fully available?

Reviewer #2: Yes

5. Is the manuscript presented in an intelligible fashion and written in standard English?

Reviewer #2: Yes

6. Review Comments to the Author

Reviewer #2: I have reviewed the revised manuscript and am pleased to report that the authors have diligently addressed all the previous comments and suggestions in a proper and comprehensive manner. The revisions have improved the clarity and quality of the article, and is now ready for publication.

Thank you for considering my review

7. PLOS authors have the option to publish the peer review history of their article (what does this mean?). If published, this will include your full peer review and any attached files.

Reviewer #2: No

---

## [Editor Report · Acceptance letter]

19 Oct 2023

PONE-D-22-27793R2 

Measuring novice-expert sense of place for a far-away place: Implications for geoscience instruction 

Dear Dr. Gold:

I'm pleased to inform you that your manuscript has been deemed suitable for publication in PLOS ONE. Congratulations! Your manuscript is now with our production department. 

Kind regards, 

on behalf of

Dr. Kendra Helen Oliver 

Academic Editor

PLOS ONE